# The role of integration host factor in biofilm and virulence of high-alcohol-producing *Klebsiella pneumoniae*

Zheng Fan,[1] Tongtong Fu,[1] Zhoufei Li,[1,2] Bing Du,[3] Xiaohu Cui,[1] Rui Zhang,[1,2] Yanling Feng,[1] Hanqing Zhao,[1] Guanhua Xue,[1] Jinghua Cui,[1] Chao Yan,[1] Lin Gan,[1] Junxia Feng,[1] Ziying Xu,[1] Zihui Yu,[1] Ziyan Tian,[1] Zanbo Ding,[1] Jinfeng Chen,[1] Yujie Chen,[1] Jing Yuan[1]

**ABSTRACT** *Klebsiella pneumoniae* is a well-known human nosocomial pathogen with an arsenal of virulence factors, including capsular polysaccharides (CPS), fimbriae, flagella, and lipopolysaccharides (LPS). Our previous study found that alcohol acted as an essential virulence factor for high-alcohol-producing *K. pneumoniae* (HiAlc *Kpn*). Integration host factor (IHF) is a nucleoid-associated protein that functions as a global virulence regulator in *Escherichia coli*. However, the regulatory role of IHF in *K. pneumoniae* remains unknown. In the present study, we found that deletion of *ihfA* or *ihfB* resulted in a slight defect in bacterial growth, a severe absence of biofilm formation and cytotoxicity, and a significant reduction in alcohol production. RNA sequencing differential gene expression analysis showed that compared with the wild-type control, the expression of many virulence factor genes was downregulated in ΔihfA and ΔihfB strains, such as those related to CPS (*rcsA*, *galF*, *wzi*, and *iscR*), LPS (*rfbABCD*), type I and type III fimbriae (*fim* and *mrk* operon), cellulose (*bcs* operon), iron transporter (*feoABC*, *fhuA*, *fhuF*, *tonB*, *exbB*, and *exbD*), quorum sensing (*lsr* operon and *sdiA*), type II secretion system (T2SS) and type VI secretion system (T6SS) (*tssG*, *hcp*, and *gspE*). Of these virulence factors, CPS, LPS, fimbriae, and cellulose are involved in biofilm formation. In addition, IHF could affect the alcohol production by regulating genes related to glucose intake (*ptsG*), pyruvate formate-lyase, alcohol dehydrogenase, and the tricarboxylic acid (TCA) cycle. Our data provided new insights into the importance of IHF in regulating the virulence of HiAlc *Kpn*.

**IMPORTANCE** *Klebsiella pneumoniae* is a well-known human nosocomial pathogen that causes various infectious diseases, including urinary tract infections, hospital-acquired pneumonia, bacteremia, and liver abscesses. Our previous studies demonstrated that HiAlc *Kpn* mediated the development of nonalcoholic fatty liver disease by producing excess endogenous alcohol *in vivo*. However, the regulators regulating the expression of genes related to metabolism, biofilm formation, and virulence of HiAlc *Kpn* remain unclear. In this study, the regulator IHF was found to positively regulate biofilm formation and many virulence factors including CPS, LPS, type I and type III fimbriae, cellulose, iron transporter, AI-2 quorum sensing, T2SS, and T6SS in HiAlc *Kpn*. Furthermore, IHF positively regulated alcohol production in HiAlc *Kpn*. Our results suggested that IHF could be a potential drug target for treating various infectious diseases caused by *K. pneumoniae*. Hence, the regulation of different virulence factors by IHF in *K. pneumoniae* requires further investigation.

**KEYWORDS** HiAlc *Kpn*, IHF, biofilm, virulence, regulate

*K*lebsiella pneumoniae is a Gram-negative human nosocomial pathogen responsible for various infections, including pneumonia, bacteremia, and liver abscesses (1–3). The bacterium comprises a variety of virulence factors, including capsule,

Address correspondence to Jing Yuan, yuanjing6216@163.com.

Zheng Fan, Tongtong Fu, and Zhoufei Li contributed equally to this article. Author order was determined on the basis of seniority.

The authors declare no conflict of interest.

See the funding table on p. 13.

lipopolysaccharides (LPS), type I and type III fimbriae, type VI secretion system (T6SS), quorum-sensing (QS) system, and siderophores, many of which are involved in biofilm formation (1, 4). Previously, we found that high-alcohol-producing *K. pneumoniae* (HiAlc *Kpn*), which was present in the intestines of 60% of patients with nonalcoholic fatty liver disease (NAFLD), was a primary causative agent of NAFLD (2). This indicated that alcohol production was also an important virulence factor for *K. pneumoniae*.

Integration host factor (IHF) belongs to nucleoid-associated proteins and was first identified as a λ-phage accessory protein involved in the site-specific recombination process (5, 6). IHF is a heterodimeric protein encoded by *ihfA* and *ihfB* which binds to sequence-specific DNA and mediates its bending, thereby facilitating the assembly of nucleoprotein structures and regulating DNA replication, recombination, repair, and other processes (7). As a global regulator, IHF plays an essential role in bacterial response to environmental stresses and pathogenesis by controlling the expression of numerous genes (8, 9). In *Escherichia coli*, IHF contributes to the formation of capsule and type I fimbriae and helps stabilize pathogenicity islands that play an important role in bacterial colonization (10–12). Moreover, IHF can promote the development of *E. coli* persisters by decreasing energy-generating components, resulting in higher levels of antibiotic tolerance (13). In *Salmonella enterica*, *ihf* deletion mediates a deficiency in biofilm formation by reducing curli fimbriae, cellulose, and pellicle production (14). In *Vibrio cholerae*, IHF positively regulates two main virulence genes, *tcpA* and *ctx*, which encode for the synthesis of toxin co-regulated fimbriae and cholera toxin (15). However, the role of IHF in *K.pneumoniae* is still unknown.

In this study, we found that deletion of *ihfA* or *ihfB* in HiAlc *Kpn* severely impaired biofilm formation and virulence and significantly reduced alcohol production. IHF positively regulated the expression of a number of virulence factors, such as capsular polysaccharides (CPS), LPS, cellulose, iron transporter, QS, type II secretion system (T2SS), T6SS, and type I and type III fimbriae. Downregulated expression of these virulence factors can explain the impaired biofilm formation in HiAlc *Kpn*. Furthermore, IHF affected the alcohol production of HiAlc *Kpn* by regulating genes related to glucose intake, the tricarboxylic acid (TCA) cycle, and fermentation. By elucidating the global regulatory role of IHF in bacterial virulence, we demonstrated that IHF could be a potential target for treating various diseases caused by *K. pneumoniae*.

## MATERIALS AND METHODS

### Bacterial strains, plasmids, and primers

The wild-type strain used in this study was high-alcohol-producing *Klebsiella pneumoniae* (W14), obtained from a patient with auto-brewery syndrome combined with NAFLD (2). Methods by Link et al. (16) and Pan et al. (17) were applied to obtain the gene deletion and complementation strains. All the strains used for culture and phenotypic analysis were grown in Luria-Bertani (LB) broth (5-g/L yeast extract, 10-g/L sodium chloride, and 10-g/L tryptone) in a shaking incubator (180–200 rpm) at 37°C. All the bacterial strains, plasmids, and primers used in this study are listed in Table S1.

### Bacterial growth curves

Growth curves of W14, Δ*ihfA*, Δ*ihfB*, Δ*ihfA*/*ihfA*, and Δ*ihfB*/*ihfB* strains were derived after subculturing in LB broth overnight. Briefly, the overnight cultures of strains were diluted 1:100 into 20 mL of fresh LB broth and grown by shaking at 180–200 rpm at 37°C. Optical density (OD)$_{600}$ measurements were performed per hour to determine the cell density. Three independent cultures were used for each assay.

### Measurement of alcohol concentration

LB cultured strains were incubated overnight, and the supernatant was taken by centrifugation at 12,000 rpm to measure alcohol concentration. The level of alcohol in

all strains was determined by headspace gas chromatography (Agilent 6850) with flame ionization detection (Headspace) in both aerobic and anaerobic conditions.

## Biofilm formation analysis

W14, $\Delta ihfA$, $\Delta ihfB$, $\Delta ihfA/ihfA$, and $\Delta ihfB/ihfB$ strains were grown overnight, diluted to $OD_{600}$ of 0.025 in LB broth, and incubated 200 µL in each well of a 96-well plate for 24 h. Subsequently, the liquid was poured from the medium, followed by washing with phosphate-buffered saline (PBS) and drying at 65°C for 15 min. Each well was stained with 1% crystal violet and washed twice with 200 µL of bleaching solution (malcohol:glacial acetic acid:$H_2O$ (vol/vol/vol) =4:1:5) per well. The medium was incubated at room temperature with gentle shaking. Finally, the biofilm quantification with crystal violet was measured at a wavelength of 590 nm.

## Observation of biofilm production by CLSM

To better analyze the effect of IHF on biofilm formation, confocal laser scanning microscope (CLSM) was used to analyze biofilm production. The cultured strains were incubated overnight and diluted at a ratio of 1:100 with fresh LB. A total of 1 mL of the diluted solution was added to a 15-mm glass bottom cell culture dish and incubated at 37°C for 48 h to form biofilm at the bottom. The culture dish was washed thrice with PBS buffer and fixed with 2.5% glutaraldehyde for 1.5 h at 4°C. After washing twice with PBS buffer, 400 µL of 50-µg/mL fluorescein isothiocyanate-conjugated concanavalin A (FITC-conA) solution was added to stain the extracellular polysaccharides at 4°C for 1 h. After washing with PBS again, the bacteria in the biofilm were observed by fluorescent staining with 10-µg/mL propidium iodide (PI) solution at 4°C for 15 min. The resulting solution was then poured and dried at room temperature. A CLSM system was used to digitize all confocal images. Fluorescent intensity, biofilm production, bacterial density, and other analyses were analyzed using ImageJ.

## Cytotoxicity assay

The cytotoxicity of *K. pneumoniae* was measured by a cell-lifting assay with subtle adjustments (18). A549 cells ($1 \times 10^5$) were inoculated into each well in a 24-well plate and cultured in Dulbecco's modified Eagle's medium (DMEM) containing 10% fetal calf serum with 5% $CO_2$ at 37°C for 24 h. A549 cells were infected with *K. pneumoniae* at a multiplicity of infection of 100 for 10 h. After that, the medium in each well was aspirated, and the remaining cells were washed with PBS twice and stained with 0.025% crystal violet (200 µL) at 37°C for 15 min. The crystal violet was discarded, and the plate was washed with 1-mL phosphate-buffered saline twice. Next, 200-µL 95% alcohol was added into each well and incubated with gentle shaking for 30 min at room temperature to dissolve the dye. Alcohol solution was used to measure absorbance at a wavelength of 490 nm.

## RNA sequencing and differential expression analysis

RNA sequencing was performed using Gene Denovo Biotechnology Co., Ltd (Guangzhou, China). Briefly, total RNA was extracted from late-exponentially growing period of W14, $\Delta ihfA$, and $\Delta ihfB$ using Trizol reagent kit (Invitrogen, Carlsbad, CA, USA), and genomic DNA was removed using recombinant DNase I. After total RNA was extracted, prokaryotic mRNA was enriched by removing rRNA by Ribo-Zero Magnetic Kit (Epicentre, Madison, WI, USA). Then, the enriched mRNA was reverse-transcribed into cDNA using NEBNext Ultra RNA Library Prep Kit for Illumina (NEB #7530; New England Biolabs, Ipswich, MA, USA). The resulting cDNA library was sequenced using Illumina Novaseq6000 by Gene Denovo Biotechnology Co. RNA differential expression analysis between two groups was performed by DESeq2 software (19). Kyoto Encyclopedia of Genes and Genomes (KEGG) and Gene Ontology (GO) enrichment analysis were conducted using KOBAS version

2.06 and Goatools. Bioinformatics analysis was performed using Omicsmart, a real-time interactive online platform for data analysis (http://www.omicsmart.com).

## RNA extraction, reverse transcription, and qRT-PCR

The bacterial suspension to be tested was first centrifuged at 12,000 rpm, and the obtained bacterial precipitate was resuspended with 1-mL trizol. Then, 0.2-mL trichloromethane was added, and the solution was vortexed for 15 s and centrifuged at 12,000 rpm for 10 min. Equal amounts of the aqueous phase were mixed lightly with isopropanol, and total RNA was extracted using the RNeasy Mini kit (Tiangen Biotech, Beijing, China). Approximately 0.5–1.0 µg of RNA was used for reverse transcription, and cDNA was synthesized by PrimeScript Reverse Transcriptase (TaKaRa, Dalian, China). The quantitative real-time PCR (qRT-PCR) experiment was performed using 20 µL of a mix including the SYBR Premix Ex TaqTM II (TaKaRa), cDNA templates, and specific forward and reverse primers. The experiment was performed using the CFX Connect Real-Time system (Bio-Rad, USA). *rpoB*, a gene encoding the DNA-directed RNA polymerase subunit β, was used as an internal reference.

## Statistical analysis

Each assay used three biological replicates to ensure accuracy. Data were statistically analyzed by GraphPad Prism (version 5.0, USA) and expressed as mean ± standard deviations. Statistically significant differences were determined by Student's *t*-test and *$P < 0.05$, **$P < 0.01$, and ***$P < 0.001$ all reflected statistical significance.

## RESULTS

### IHF slightly influenced *in vitro* growth in HiAlc *Kpn*

IHF typically comprises two subunits, IHFα and IHFβ, encoded by *ihfA* and *ihfB*, respectively (20, 21). To characterize the function of IHF in the pathogenicity of HiAlc *Kpn*, we generated two mutants: Δ*ihfA* and Δ*ihfB*. Compared with wild-type strain W14, the growth curves of Δ*ihfA* and Δ*ihfB* in LB broth demonstrated a slightly slower growth, and compensation of *ihfA* and *ihfB* restored bacterial growth (Fig. 1A). However, no difference in growth was observed among wild-type strain W14, Δ*ihfA*, and Δ*ihfB* mutant strains on the solid plates (Fig. 1B). These results indicated that the deletion of IHF slightly affected the growth of HiAlc *Kpn*.

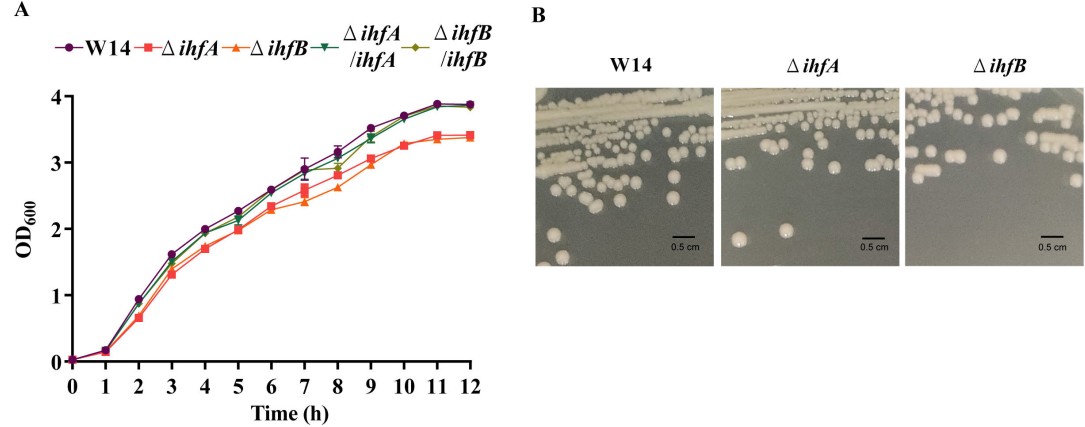

**FIG 1** Deletion of *ihfA* or *ihfB* slightly influenced the growth of HiAlc *Kpn* in LB broth. (A) The growth curves of W14, Δ*ihfA*, Δ*ihfB*, Δ*ihfA/ihfA*, and Δ*ihfB/ihfB* were measured by optical density ($OD_{600}$) per hour over a period of 12 h. (B) Typical colony images of W14, Δ*ihfA*, and Δ*ihfB*.

## IHF affected biofilm formation and cytotoxicity in HiAlc *Kpn*

To examine whether IHF affected biofilm formation in HiAlc *Kpn*, we performed Crystal violet (CV) staining and CLSM. Deletion of *ihfA* or *ihfB* significantly impaired the ability of biofilm formation in HiAlc *Kpn*. Compensation with *ihfA* or *ihfB* restored biofilm formation to the level of wild-type strain W14 (Fig. 2A through C). Meanwhile, we tested bacterial cytotoxicity by measuring their ability to detach A549 cells from the culture plates. Compared to wild-type strain W14, *ihfA* and *ihfB* deletion exhibited decreased cytotoxicity. Complementation of *ihfA* or *ihfB* restored the bacterial cytotoxicity (Fig. 2D). Our results demonstrated that IHF affected the biofilm formation and bacterial virulence, similar to the effects observed in *Salmonella enterica* and *E. coli* (10, 21, 22).

## IHF contributed to alcohol production in HiAlc *Kpn*

Our previous study found that HiAlc *Kpn* could produce large amounts of endogenous alcohol, which led to NAFLD, indicating that alcohol might be a critical pathogenic virulence factor in HiAlc *Kpn* (2). As shown in Fig. 3, alcohol production significantly decreased in Δ*ihfA* and Δ*ihfB* under both aerobic and anaerobic conditions, while compensation with *ihfA* and *ihfB* restored alcohol production. These results showed that IHF was required for alcohol production in HiAlc *Kpn*.

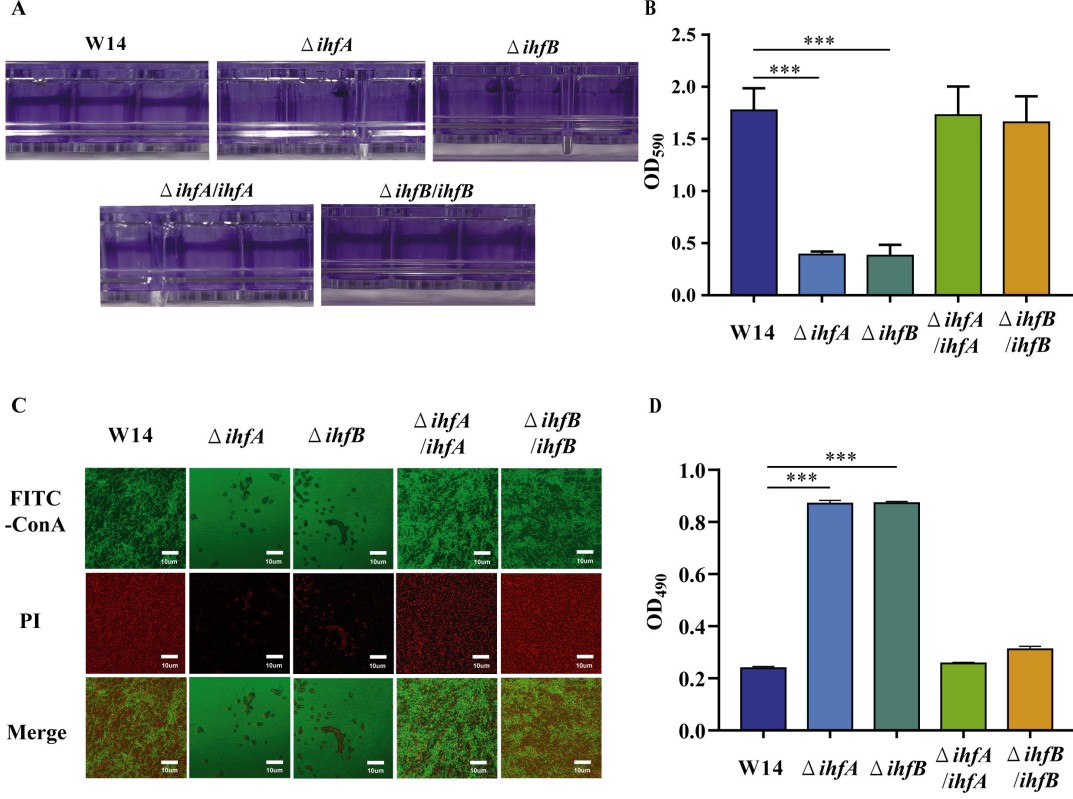

**FIG 2** Deletion of *ihfA* or *ihfB* decreased the biofilm formation and virulence in HiAlc *Kpn*. (A) Images of biofilm formed by W14, Δ*ihfA*, Δ*ihfB*, Δ*ihfA/ihfA*, and Δ*ihfB/ihfB* were stained with 1% CV and adhered on plastic centrifuge tubes. (B) Biofilm was stained with 1% CV. The extracted color was dissolved with 33% bleaching solution and measured at $OD_{590}$. ***$P < 0.001$, compared to wild-type strain W14 by Student's *t*-test. (C) CLSM of biofilm formation in W14, Δ*ihfA*, Δ*ihfB*, Δ*ihfA/ihfA*, and Δ*ihfB/ihfB* was observed after incubation for 48 h. (D) Cytotoxicity of W14, Δ*ihfA*, Δ*ihfB*, Δ*ihfA/ihfA*, and Δ*ihfB/ihfB*. A549 cells were infected with the indicated strains at a MOI of 100. After 10 h of infection and 15-min crystal violet staining, cells attached to the plate were measured at $OD_{490}$. ***$P < 0.001$, compared to wild-type strain W14 by Student's *t*-test.

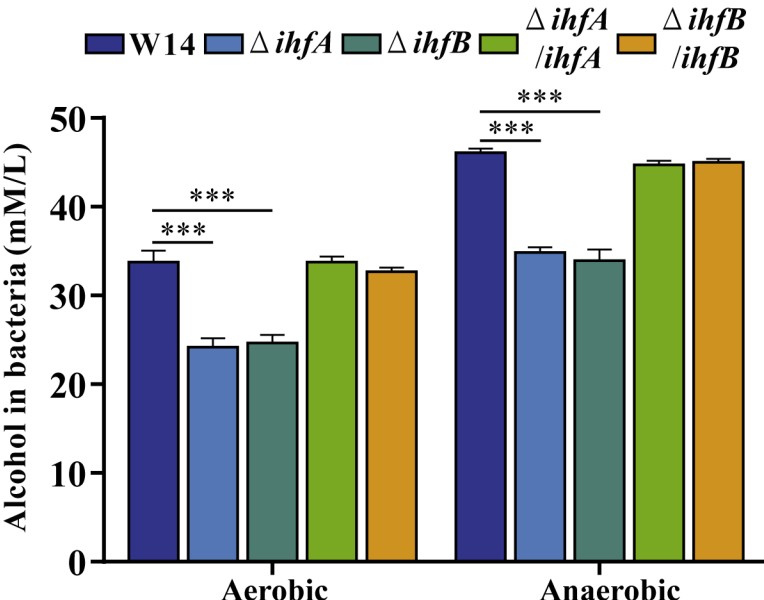

**FIG 3** Deletion of *ihfA* or *ihfB* decreased the alcohol production in HiAlc *Kpn*. The alcohol-producing ability of W14, ΔihfA, ΔihfB, ΔihfA/ihfA, and ΔihfB/ihfB was measured in aerobic and anaerobic conditions. ***$P < 0.001$, compared to wild-type strain W14 by Student's t-test.

## Profiling gene expression of Δ*ihfA* and Δ*ihfB*

To understand the mechanism of IHF-mediated regulation of bacterial biofilm and virulence, we examined the gene expression profiles of the late-exponentially growing W14, Δ*ihfA*, and Δ*ihfB*. Differentially expressed genes (DEGs) with $P < 0.05$ and |log2 (fold change)| of >1 were identified by comparing the RNA-sequencing (RNA-seq) data with that of wild-type strain W14. Compared with wild-type strain W14, the number of DEGs in Δ*ihfA* and Δ*ihfB* strains was 700 (320 upregulated and 380 downregulated genes) and 689 (315 upregulated and 374 downregulated genes), respectively. The number of DEGs between Δ*ihfA* and Δ*ihfB* was only six (five upregulated and one downregulated genes), which suggested that *ihfA and ihfB* might play similar regulatory roles in HiAlc *Kpn* (Fig. 4A and B). The DEGs of Δ*ihfA* and Δ*ihfB* strains were analyzed using GO enrichment. A total of 20 significantly enriched GO terms from three categories (biological process, molecular function, and cellular component) are shown in Fig. 4C and D. Most enriched DEGs among GO terms corresponded to cellular process, metabolic process, catalytic activity, binding, and cellular anatomical entity. Based on the KEGG pathway analysis, the upregulated genes were enriched in metabolic pathways, microbial metabolism in diverse environments, and ABC transporters pathways (Fig. 4E and F). Notably, the downregulated DEGs in Δ*ihfA* and Δ*ihfB* strains relative to wild-type strain W14 were enriched in the biofilm information, quorum-sensing pathways, suggesting that IHF can play an important role in biofilms and virulence of *K. pneumoniae* (Fig. 4G and H).

## IHF-regulated genes related to bacterial biofilm and virulence

CPS are closely related to biofilm formation and bacterial virulence (23). RcsA plays an important role in Rcs phosphorelay system, which regulates CPS and biofilm formation in *K. pneumoniae* (24). When RcsA is inhibited, CPS expression is significantly decreased (25). The genes *galF* and *wzi* are involved in capsule formation, coding for UDP-glucose pyrophosphorylase and an outer membrane protein, respectively (26). RcsA participates in the CRP-cAMP-mediated regulation of *galF* transcription and influences CPS biosynthesis in *K. pneumoniae*. IscR, an Fe-S cluster-containing transcription factor, plays a key role in CPS synthesis and iron acquisition in *K. pneumoniae* (4, 27). RNA-seq and qRT-PCR

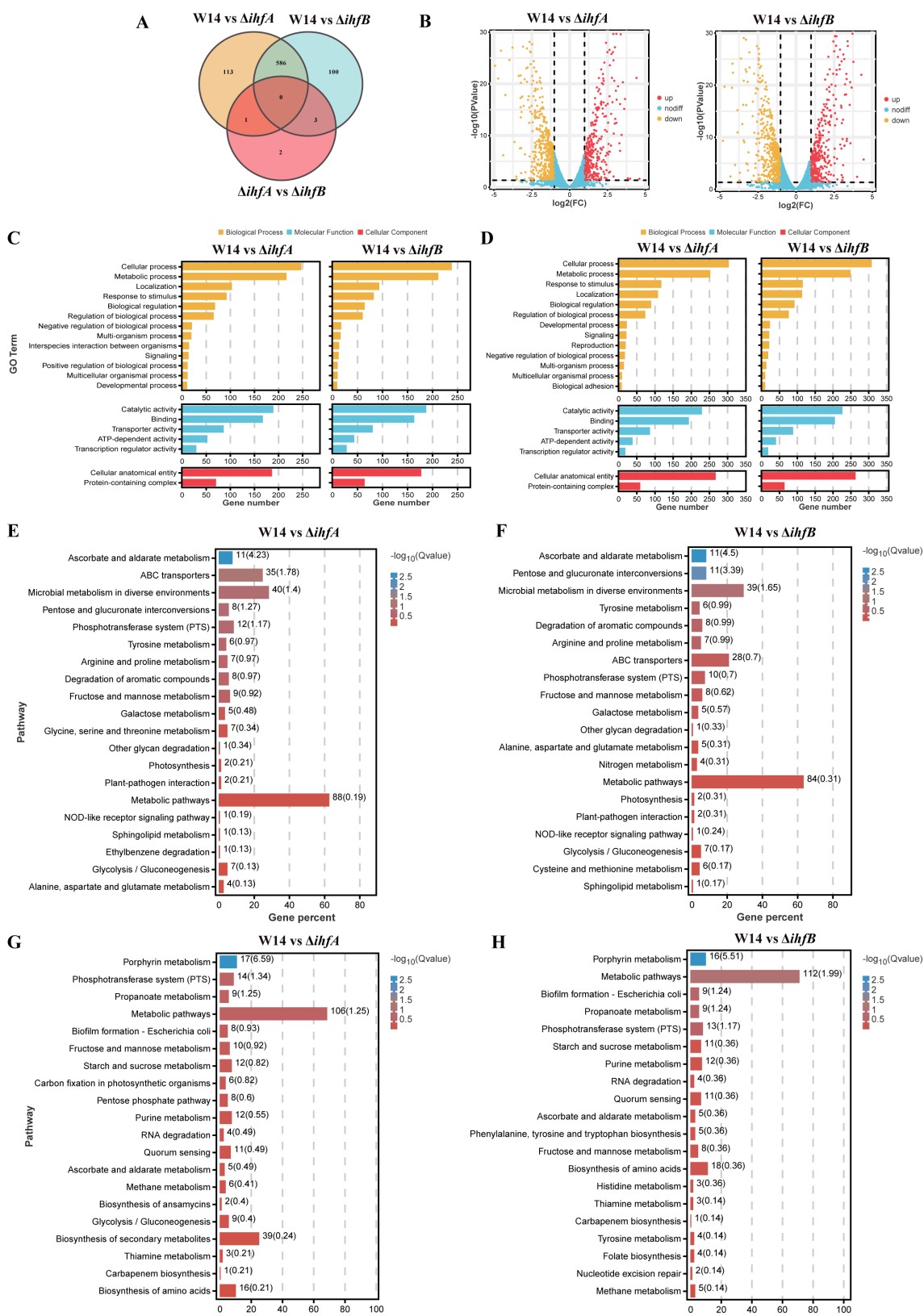

**FIG 4** Profiling gene expression of Δ*ihfA* and Δ*ihfB*. Analysis of total RNA sequencing. (A) The Venn diagram shows the overlapped DEGs numbers of W14 and Δ*ihfA*, W14 and Δ*ihfB*, Δ*ihfA* and Δ*ihfB*. (B) Volcano plot of DEGs between W14 and Δ*ihfA* or Δ*ihfB*. GO enrichment analysis of upregulated (C) and downregulated (D) DEGs in Δ*ihfA* and Δ*ihfB*. KEGG pathway enrichment analysis of upregulated DEGs in Δ*ihfA* (E) and Δ*ihfB* (F). KEGG pathway enrichment analysis of downregulated DEGs in Δ*ihfA* (G) and Δ*ihfB* (H).

results indicated that the expression of *rcsA*, *galF*, *wzi*, and *iscR* was significantly lower in ΔihfA and ΔihfB and showed that IHF could positively regulate the synthesis of CPS in HiAlc *Kpn* (Fig. 5 and 6A).

LPS, cellulose, type I fimbriae, and type III fimbriae are also involved in *K. pneumoniae* biofilm formation. *wzm* (*rfbA*) and *wbbM* (*rfbC*) belong to the *rfb* gene cluster, encoding enzymes for the biosynthesis of O-antigen in *K. pneumoniae* (4, 28). Cellulose is an important component of bacterial biofilm. Bacterial cellulose synthesis and translocation are regulated by the inner membrane-associated *bcsABZC* and *bcsEFG* operons in *E. coli* and *Salmonella* (29, 30). *bcsA*, *bcsB*, and *bcsC* encode a cellulose synthase enzyme, c-di-GMP-binding protein, and cellulose oxidoreductase enzyme, respectively (30). Type I and type III fimbriae are important virulence factors affecting biofilm formation, cell attachment, and pathogenicity in *K. pneumoniae*, which are encoded by *fim* and *mrk* operons, respectively (31, 32). RNA-seq differential gene expression analysis and qRT-PCR showed that the expression of genes related to LPS (*rfbABCD*) (Fig. 6B), cellulose (*bcsABCZ*, *bcsEFG*, *bcsO*, and *bcsQ*) (Fig. 6C), and type I and type III fimbriae (*fimABC-DEFGH*, *mrkAB*, and *mrkHIJ*) (Fig. 6D and E) was significantly downregulated compared to that in wild-type control (Fig. 5).

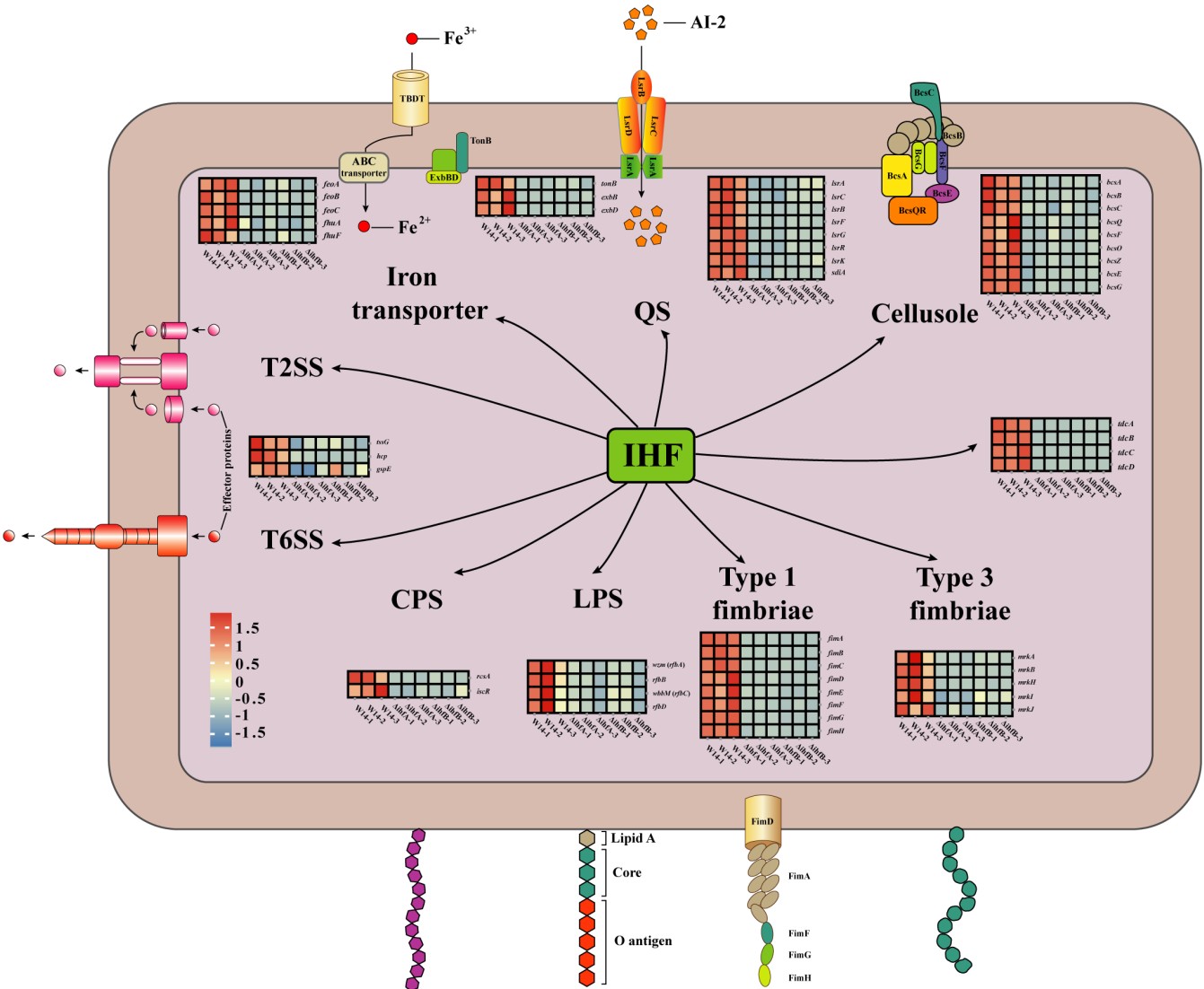

**FIG 5** Regulatory network of IHF in HiAlc *Kpn*. The regulatory network of IHF in HiAlc *Kpn*. IHF positively regulates the synthesis of CPS, LPS, cellulose, type I and type III fimbriae, T2SS, T6SS, iron transporter and QS by IHF. Gene expression measured by RNA sequencing is shown by heat map.

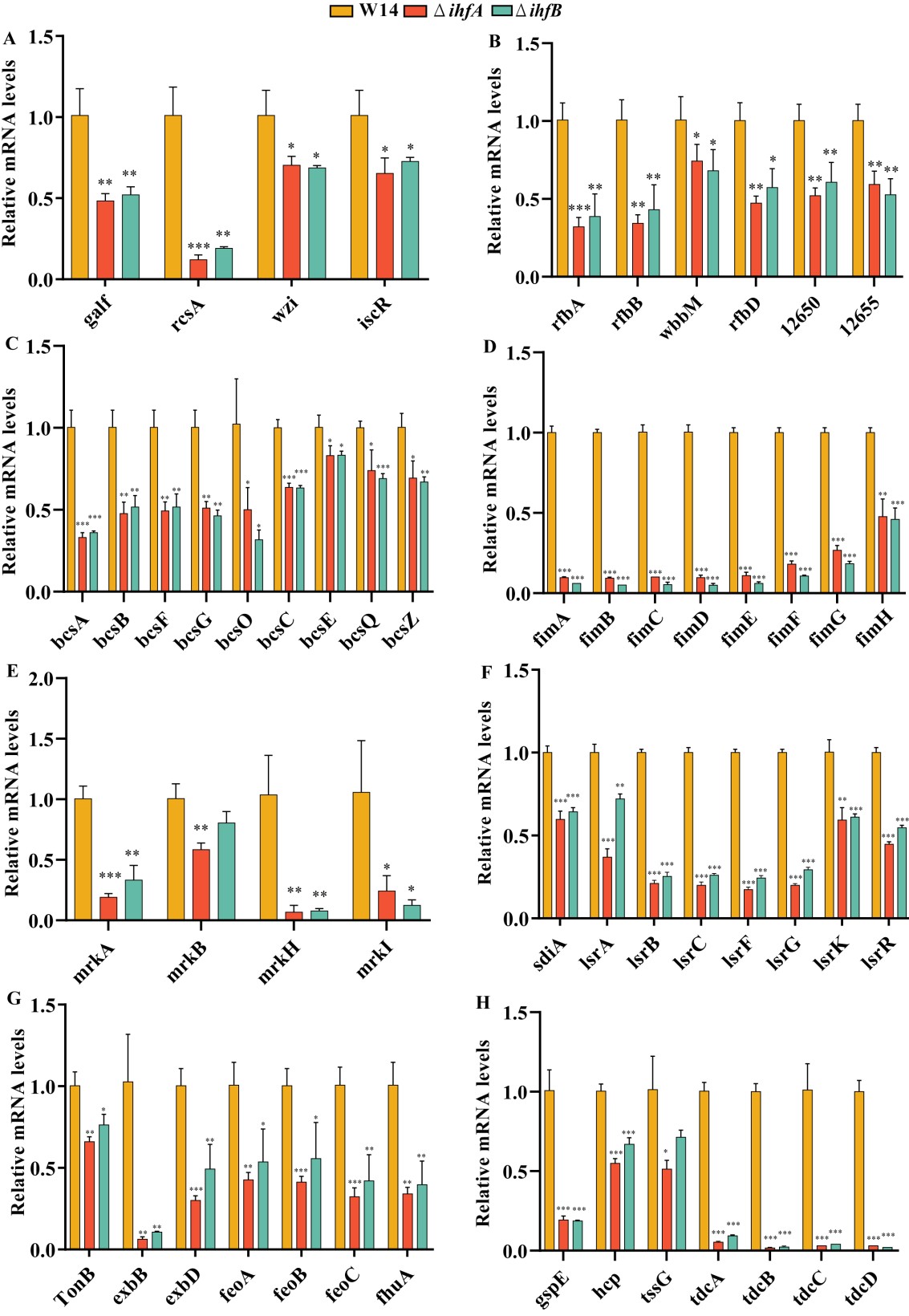

**FIG 6** Deletion of *ihfA* or *ihfB* decreased the expressions of biofilm and virulence-related genes. The relative mRNA levels of W14, Δ*ihfA*, and Δ*ihfB* in CPS (A), LPS (B), cellulose (C), type I fimbriae (D), type III fimbriae (E), QS system (F), iron acquisition (G), and other virulence-related genes (H) were calculated by qRT-PCR. *$P <$ 0.05, **$P <$ 0.01, ***$P <$ 0.001, compared to wild-type strain W14 by Student's *t*-test.

LuxS/AI-2 QS system can detect cell density and affect the virulence of *K. pneumoniae* (33, 34). AI-2 is produced by the enzyme LuxS and consumed intracellularly by the Lsr system, which consists of the ATP-binding cassette transporter LsrACDB, AI-2 modifier protein LsrFG, AI-2 uptake repressor LsrR, and AI-2 phosphorylation kinase LsrK (35). SdiA, a LuxR-type regulator, negatively regulates fimbriae expression, biofilm synthesis, and AI-2 production in *K. pneumoniae* (34). In this study, the expression of *lsrACB*, *lsrFG*, *lsrRK*, and *sdiA* was significantly downregulated in *ihfA* and *ihfB* deletion strains (Fig. 5 and 6F).

Iron acquisition is vital for bacterial survival and growth. Its absorption depends on a specific outer membrane transporter, in which TonB-dependent transporters (TBDTs) can efficiently bind to the iron-containing substrates. After binding to the substrates, the conformation of TBDTs changes, followed by interaction with TonB via the N-terminal conserved region to obtain energy to complete the transport (36). After passing through the outer membrane, the iron-containing substrate binds to a periplasmic cavity-binding protein. It then enters the cytoplasm via the Feo transport system (including FeoABC) on the inner membrane. The Feo system is crucial for the colonization and virulence of pathogens (37). FhuA is a multifunctional protein acting as a transmembrane receptor in the outer membrane that recognizes iron carriers and enters the periplasm in a TonB-dependent manner (38). FhuF is a member of the iron carrier reductase (FSR) subfamily (39). It participates in the removal of iron carriers (40). Our results showed that iron transport-related gene expression (*feoABC*, *fhuA*, *fhuF*, *tonB*, *exbB*, and *exbD*) was significantly decreased in *ihfA* and *ihfB* deletion strains (Fig. 5 and 6G).

Secretion systems, from type I secretion system to T6SS, are regulated by IHF in *Dickeya zeae* (41). In HiAlc *Kpn*, we found that the expression of genes related to T2SS and T6SS (*tssG*, *hcp*, and *gspE*) was significantly decreased in Δ*ihfA* and Δ*ihfB* mutant strains (Fig. 5 and 6H). Type II secretion system is a sophisticated multiprotein machinery essential for bacterial pathogenicity (42, 43). GspE is an ATPase belonging to the type II/type IV secretion family. The energy for T2SS is generated by GspE-mediated ATP hydrolysis to transport folded proteins outside the cell through the GspD outer membrane channel (44). T6SS is a contact-dependent protein secretion apparatus involved in multiple processes related to bacterial virulence. Bacteria use T6SS to compete with the host and to coordinate the invasion process (45–47). T6SS needle consists of an Hcp hexamer ring, and TssG is a protein that helps to localize the needle in the donor cell (48).

The operon *tdcABCDEFG*, which consists of the regulatory gene *tdcA* and the structural gene *tdcBCDEFG*, is involved in the transport and metabolism of L-threonine and L-serine during anaerobic growth in *E. coli* (49). Research indicated that Tdc affects the intestinal colonization of *K. pneumoniae* (50). Our data showed that the expression of *tdcABCD* was significantly downregulated in both Δ*ihfA* and Δ*ihfB* (Fig. 5 and 6H).

## IHF-regulated genes related to the TCA cycle and fermentation

HiAlc *Kpn* can mediate the induction of NAFLD by producing excess endogenous alcohol using glucose as the main carbon source. EIICB$^{glc}$, a major glucose transporter, is encoded by *ptsG*. Glucose that enters the cells is converted to pyruvate via glycolysis (51). The generation of acetyl-coenzyme A (acetyl-CoA) from pyruvate is mediated by pyruvate dehydrogenase complex and pyruvate formate-lyase (PFL) in two main ways. Acetyl-CoA can enter the TCA cycle or be catalyzed by alcohol dehydrogenase (*adhE*) to convert acetaldehyde to produce alcohol (52–55) (Fig. 7A). In *K. pneumoniae*, pyruvate formate-lyase (*pflB*) and *adhE* encodes PFL and the main alcohol dehydrogenase, respectively. qRT-PCR results showed that *ptsG*, *pflB*, and *adhE* expression decreased significantly in Δ*ihfA* and Δ*ihfB* under high glucose condition (Fig. 7B). Overexpression of *adhE* significantly increased alcohol production in the Δ*ihfA* and Δ*ihfB* mutant strains (Fig. 7C). Meanwhile, the expression of genes related to the TCA cycle (*gltA*, *icd*, *sucABCD*, *sdhB*, and *fumA*) increased significantly (Fig. 7D). These results indicated that IHF was required

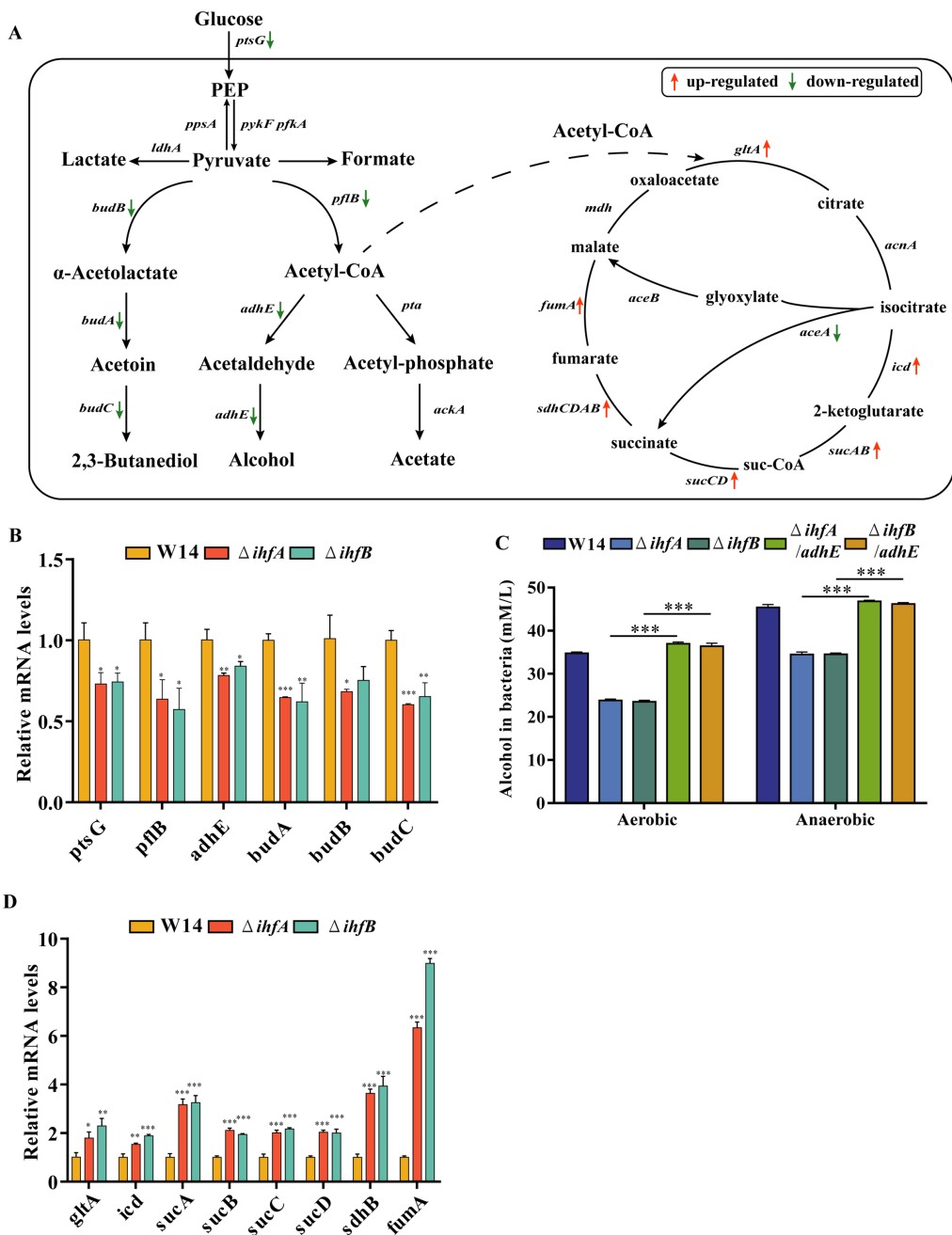

**FIG 7** IHF-regulated genes related to the TCA cycle and fermentation. (A) Metabolic pathways and related genes of HiAlc *Kpn*. The picture shows up- and downregulated genes in Δ*ihfA* and Δ*ihfB* compared with wild-type strain W14. (B) The relative mRNA levels of fermentation-related genes in W14, Δ*ihfA*, and Δ*ihfB* were calculated by qRT-PCR. (C) The alcohol-producing ability of W14, Δ*ihfA*, Δ*ihfB*, Δ*ihfA/adhE*, and Δ*ihfB/adhE* was measured in aerobic and anaerobic conditions. (D) The relative mRNA levels of TCA cycle-related genes in W14, Δ*ihfA*, and Δ*ihfB* were calculated by qRT-PCR. *$P < 0.05$; **$P < 0.01$, ***$P < 0.001$, compared to wild-type strain W14 by Student's *t*-test.

for glucose intake and regulated the expression of TCA and fermentation-related genes, thereby affecting the production of alcohol.

## DISCUSSION

Our previous study found that HiAlc *Kpn* is associated with nearly 60% of individuals with NAFLD in a Chinese cohort (2). The large amount of endogenous alcohol

produced by intestinal bacteria can lead to hepatic steatosis, mitochondrial dysfunction, and inflammatory response, and the long-term accumulation of these pathological changes can predispose an individual to the development of NAFLD (2). Furthermore, the overgrowth of LPS-producing pathogenic strains with high endotoxin activity including *K. pneumoniae* is related to the onset of NAFLD (56, 57). LPS acts as a pathogen-associated molecular pattern that binds to toll-like recepter 4 and triggers an essential inflammatory cascade during the progression of NAFLD (57, 58). Therefore, both high-alcohol and high-endotoxin activities of bacteria are important for the development of NAFLD. This study demonstrated that IHF can positively regulate alcohol production and LPS in HiAlc *Kpn*. Therefore, IHF may be a potential drug target for treating NAFLD caused by HiAlc *Kpn*.

As a regulator, IHF binds to DNA by recognizing consensus sequences 5′‐WAT-CAANNNNTTR‐3′ (where W = A or T, N = any base, and R = A or G) in *E. coli* (59, 60). The consensus sequence was used to predict potential sites for binding IHF to DNA in HiAlc *Kpn W14* using FIMO version 5.5.1 (61). Among the genes of HiAlc *Kpn* regulated by IHF at the transcriptional level, IHF-binding sites were found in the promoter regions of genes *fimA*, *mrkH*, and *tdcA*. However, the precise IHF-binding sites in *K. pneumoniae* still need to be further identified, which will deepen the understanding of the regulatory mechanism of IHF.

The function of IHF has been explored in several bacterial species. In *E.coli*, the expression of type I pili, group 2 capsule, and *tdc* operon and the colonization of the urinary tract can be affected by IHF (10, 12, 62, 63). In *S. enterica*, IHF positively regulated the biofilm formation by affecting the curli fimbriae expression, cellulose production, and pellicle formation (10, 12). In *Vibrio fluvialis*, IHF bound to the promoter of T6SS major cluster to regulate the expression and secretion of Hcp (64). Consistent with the previous research, our results indicated that in *K. pneumoniae*, IHF positively regulated genes related to cellulose synthesis (*bcs* operon), CPS (*rcsA*, *galF*, *wzi*, and *iscR*), type I and type III fimbriae (*fim* and *mrk* operons), T2SS and T6SS (*gspE*, *hcp*, and *tssG*), LPS (*rfbABCD*), and *tdc* operon. These results indicated that the regulatory function of IHF is relatively conserved among different strains.

It is worth noting that IHF positively regulates *lsrACDBFG* operon and negatively regulates *lsrRK* in *Aggregatibacter actinomycetemcomitans*, which differs from the findings presented here (65). Our RNA-seq differential gene expression analysis and qRT-PCR demonstrated that both *lsrABCFG* and *lsrRK* were positively regulated by IHF (Fig. 5 and 6F). The function of *lsrRK* operon and its regulation by IHF are worth investigating. Furthermore, we found that IHF was a positive regulator of iron transporters including TBDTs, Feo transport system, and ExbB-ExbD-TonB protein complex, which has not yet been reported. In summary, IHF acts as a global regulator affecting a wide range of virulence factors. There may still be unknown virulence factors regulated by IHF, and further mechanisms should be explored.

## ACKNOWLEDGMENTS

We would like to thank Editage (www.editage.cn) for English language editing.

This work was supported by grants from the National Natural Science Foundation for Key Programs of China Grants (82130065) and National Natural Science Foundation of China (32200159 and 82272352).

J.Y., Z.F., and T.F. designed the experiments. Z.F., T.F., Z.L., B.D., X.C., R.Z., Y.F., and H.Z. performed the experiments. The other authors analyzed the results. Z.F. and Z.L. wrote the manuscript. J.Y. and Z.F. revised the manuscript. All authors reviewed the manuscript.

## AUTHOR AFFILIATIONS

[1]Department of Bacteriology, Capital Institute of Pediatrics, Beijing, China
[2]Graduate School of Peking Union Medical College, Beijing, China
[3]University of Edinburgh, Edinburgh, United Kingdom

## AUTHOR ORCIDs

Jing Yuan  http://orcid.org/0000-0002-6939-9676

## FUNDING

| Funder | Grant(s) | Author(s) |
|---|---|---|
| MOST \| National Natural Science Foundation of China (NSFC) | 82130065 | Jing Yuan |
| MOST \| National Natural Science Foundation of China (NSFC) | 32200159 | Zheng Fan |
| MOST \| National Natural Science Foundation of China (NSFC) | 82272352 | Jinghua Cui |

## AUTHOR CONTRIBUTIONS

Zheng Fan, Funding acquisition, Investigation, Methodology, Software, Writing – original draft, Writing – review and editing | Tongtong Fu, Formal analysis, Investigation, Methodology, Software, Writing – original draft, Writing – review and editing | Zhoufei Li, Investigation, Resources, Writing – original draft, Writing – review and editing | Bing Du, Formal analysis, Methodology | Xiaohu Cui, Resources, Software | Rui Zhang, Project administration, Resources | Yanling Feng, Project administration, Software | Hanqing Zhao, Validation, Visualization | Guanhua Xue, Data curation, Formal analysis | Jinghua Cui, Conceptualization, Methodology | Chao Yan, Project administration, Validation | Lin Gan, Formal analysis, Resources | Junxia Feng, Methodology, Validation | Ziying Xu, Data curation, Formal analysis | Zihui Yu, Software, Validation | Ziyan Tian, Methodology, Resources | Zanbo Ding, Investigation, Resources | Jinfeng Chen, Funding acquisition, Investigation | Yujie Chen, Resources, Supervision | Jing Yuan, Conceptualization, Funding acquisition, Investigation, Writing – review and editing

## DATA AVAILABILITY

The raw sequence data of RNA sequencing in this study have been deposited in the Genome Sequence Archive in National Genomics Data Center, Beijing Institute of Genomics (China National Center for Bioinformation), Chinese Academy of Sciences, under accession number CRA010192.

## ADDITIONAL FILES

The following material is available online.

### Supplemental Material

**Table S1 (Spectrum01170-23-s0002.pdf).** Bacterial strains, plasmids, and primers used in this study.

### Open Peer Review

**PEER REVIEW HISTORY (review-history.pdf).** An accounting of the reviewer comments and feedback.

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
