## [Reviewer comments · Microbiology Spectrum]

Microbiology Spectrum

The Role of Integration Host Factor in Biofilm and Virulence of High-Alcohol-Producing *Klebsiella pneumoniae*

Zheng Fan, Tongtong Fu, Zhoufei Li, Bing Du, Xiaohu Cui, Rui Zhang, Yanling Feng, Hanqing Zhao, Guanhua Xue, Jinghua Cui, Chao Yan, Lin Gan, Junxia Feng, Ziyang Xu, Zihui Yu, Ziyang Tian, Zanbo Ding, Jinfeng Chen, Yujie Chen, and Jing Yuan

Corresponding Author(s): Jing Yuan, Capital Institute of Pediatrics

Review Timeline:

Submission Date:	March 20, 2023
Editorial Decision:	May 25, 2023
Revision Received:	June 21, 2023
Accepted:	July 28, 2023

Editor: Fei Chen

Reviewer(s): The reviewers have opted to remain anonymous.

Transaction Report:

DOI: <https://doi.org/10.1128/spectrum.01170-23>

May 25, 2023

Prof. Jing Yuan
Capital Institute of Pediatrics
Department of Bacteriology
No. 2 yabao road, Chaoyang District
Beijing, Beijing 100020
China

Re: Spectrum01170-23 (**The Role of Integration Host Factor in Biofilm and Virulence of High-Alcohol-Producing *Klebsiella pneumoniae***)

Dear Prof. Jing Yuan:

While this study holds a certain level of significance, the article still presents a few scientific and writing issues that require improvement. When submitting the revised version of your paper, please provide (1) point-by-point responses to the issues raised by the reviewers as file type "Response to Reviewers," not in your cover letter, and (2) a PDF file that indicates the changes from the original submission (by highlighting or underlining the changes) as file type "Marked Up Manuscript - For Review Only". Please use this link to submit your revised manuscript - we strongly recommend that you submit your paper within the next 60 days or reach out to me. Detailed instructions on submitting your revised paper are below.

Link Not Available

Sincerely,

Fei Chen

Journals Department
Reviewer comments:

Reviewer #1 (Comments for the Author):

The manuscript presents the study of the effects of IHF in a high-alcohol-producing *K. pneumoniae* strain W14. It's very interesting to find that the deletion of ihfA/B had an effect on the biofilm and virulence. Apart from these aspects, many formulations and statements are inadequate.

1. there exist different representations of wild type, wild type strain W14, wild type strain, and WT, please use the same expression.
2. L196/L260: " Δ ihfA and Δ ihfB deletion" should be "ihfA and ihfB deletion"

3. "IhfA" or "ihfA" should be written in a standard format ?
4. Δ ihfB groups or strains or Δ ihfB ?
5. there exists more than one space in some words.
6. L225: "in KEGG pathways" should be deleted.
7. The legends of Figure 1,2,3 are the same expression as the titles in the results. Line 619 Figure 7 B and D should be specifically referred to the related gene sets. Please clarify the grouping and corresponding classification names of these genes, such as the legend of the subgraph in Figure 6 could be revised as "The relative mRNA levels of W14, Δ ihfA, and Δ ihfB in CPS (A), LPS (B)...by qRT-PCR."

Reviewer #2 (Comments for the Author):

1. The paper is mainly focused on the regulatory role of IHF in *Klebsiella pneumoniae*, and Why did the authors choose high-alcohol-producing *Klebsiella pneumoniae* instead of hypervirulent *Klebsiella pneumoniae*?
2. How many samples are in each the RNAseq? From Figure 5, there are only two samples in each group.
3. In the results of RNAseq (Figure 4B and C), it is not specified which pathways are up-regulated/down-regulated by Δ ihfA and Δ ihfB.

Staff Comments:

Preparing Revision Guidelines

Please return the manuscript within 60 days; if you cannot complete the modification within this time period, please contact me. If you do not wish to modify the manuscript and prefer to submit it to another journal, please notify me of your decision immediately so that the manuscript may be formally withdrawn from consideration by Microbiology Spectrum.

Dear Microbiology spectrum editors and reviewers,

We appreciate the time and efforts by the editor and reviewers of 'Microbiology spectrum' in reviewing this manuscript. In response to the reviewer's remarks, we have revised our manuscript meticulously to our capacity and tried to incorporate all the suggestions made by the reviewers. We hope that the revised version could meet the publication requirements of Microbiology spectrum. And the modified/corrected parts are mentioned according to the line numbers.

A point-by-point response to Reviewer #1 comments:

1. There exist different representations of wild type, wild type strain W14, wild type strain, and WT, please use the same expression.

Response:

Thank you very much for your suggestion. We have changed different types of expression to a uniform expression "wild type strain W14".

2. L196/L260: " $\Delta ihfA$ and $\Delta ihfB$ deletion" should be "*ihfA* and *ihfB* deletion"

Response:

Thank you very much for your suggestion. We have changed " $\Delta ihfA$ and $\Delta ihfB$ deletion" to "*ihfA* and *ihfB* deletion". Please refer to line 197 and line 264 for the changes.

3. "IhfA" or "ihfA" should be written in a standard format?

Response:

Thank you very much for your suggestion. We have rewritten "IhfA" to "*ihfA*". Please refer to line 218 for the changes.

4. $\Delta ihfB$ groups or strains or $\Delta ihfB$?

Response:

Thank you very much for your suggestion. We have changed " $\Delta ihfA$ and $\Delta ihfB$ groups" to " $\Delta ihfA$ and $\Delta ihfB$ strains". Please refer to line 215 and line 219 for the changes.

5. there exists more than one space in some words.

Response:

Thank you very much for your suggestion. We have deleted the excess space.

6. L225: "in KEGG pathways" should be deleted.

Response:

Thank you very much for your suggestion. We have deleted "in KEGG pathways".

7. The legends of Figure1,2,3 are the same expression as the titles in the results.

Line619 Figure7 B and D should be specifically referred to the related gene sets.

Please clarify the grouping and corresponding classification names of these genes, such as the legend of the subgraph in Figure 6 could be revised as "The relative mRNA levels of W14, $\Delta ihfA$, and $\Delta ihfB$ in CPS (A), LPS (B)...by qRT-PCR."

Response:

Thank you very much for your suggestions.

i) We have changed the titles of Figure 1-3.

ii) We have changed the relevant description. Please refer to line 624-625 and line 628-629 for the changes.

iii) We have rewritten this description to " The relative mRNA levels of W14, $\Delta ihfA$ and $\Delta ihfB$ in CPS(A), LPS(B), cellulose(C), type 1 fimbriae(D), type 3 fimbriae(E), QS system (F), iron acquisition(G) and others virulence related genes(H) were calculated by qRT-PCR". Please refer to line 617-619 for the changes.

A point-by-point response to Reviewer #2 comments:

1. The paper is mainly focused on the regulatory role of IHF in Klebsiella

pneumoniae, and Why did the authors choose high-alcohol-producing *Klebsiella pneumoniae* instead of hypervirulent *Klebsiella pneumoniae*?

Response:

Thank you for the inquiry. Our previous studies have demonstrated that high-alcohol-producing *K. pneumoniae* (HiAlc *Kpn*) mediated the development of NAFLD by producing excess endogenous alcohol in vivo. In this study, we explored the regulatory functions of IHF, including the ability of bacteria to produce alcohol. Results revealed that IHF could positively regulate bacterial alcohol production and indicated that IHF could be a potential drug target for the treatment of NAFLD caused by HiAlc *Kpn*. I hope this could explain the reason that we choose high-alcohol-producing *Klebsiella pneumoniae* instead of hypervirulent *Klebsiella pneumoniae*.

2. How many samples are in each the RNAseq? From Figure 5, there are only two samples in each group.

Response:

Thank you for indicating that. There are three samples in each the RNAseq. We have made changes to the Figure 5.

3. In the results of RNAseq (Figure 4B and C), it is not specified which pathways are up-regulated/down-regulated by $\Delta ihfA$ and $\Delta ihfB$.

Response:

Thank you very much for your suggestion. We presented the GO and KEGG pathways analyses results of up-regulated and down-regulated genes respectively. Please refer to line 220-229 and Figure 4 for the changes.

July 28, 2023

Prof. Jing Yuan
Capital Institute of Pediatrics
Department of Bacteriology
No. 2 yabao road, Chaoyang District
Beijing, Beijing 100020
China

Re: Spectrum01170-23R1 (**The Role of Integration Host Factor in Biofilm and Virulence of High-Alcohol-Producing *Klebsiella pneumoniae***)

Dear Prof. Jing Yuan:

Your manuscript has been accepted, and I am forwarding it to the ASM Journals Department for publication. You will be notified when your proofs are ready to be viewed.

Sincerely,

Fei Chen
Editor, Microbiology Spectrum
